# Assembly mechanism of a Tad secretion system secretin-pilotin complex

Matteo Tassinari [1,3], Marta Rudzite [2], Alain Filloux [2,4] & Harry H. Low [1] ✉

The bacterial Tight adherence Secretion System (TadSS) assembles surface pili that drive cell adherence, biofilm formation and bacterial predation. The structure and mechanism of the TadSS is mostly unknown. This includes characterisation of the outer membrane secretin through which the pilus is channelled and recruitment of its pilotin. Here we investigate RcpA and TadD lipoprotein from *Pseudomonas aeruginosa*. Light microscopy reveals RcpA colocalising with TadD in *P. aeruginosa* and when heterologously expressed in *Escherichia coli*. We use cryogenic electron microscopy to determine how RcpA and TadD assemble a secretin channel with C13 and C14 symmetries. Despite low sequence homology, we show that TadD shares a similar fold to the type 4 pilus system pilotin PilF. We establish that the C-terminal four residues of RcpA bind TadD - an interaction essential for secretin formation. The binding mechanism between RcpA and TadD appears distinct from known secretin-pilotin pairings in other secretion systems.

The Tad secretion system (TadSS) system was originally identified in *Aggregatibacter actinomycetemcomitans*, the cause of localised aggressive periodontitis, where it secretes surface pili that vigorously promote cell adherence and biofilm formation[1]. Other pathogens where the TadSS exists or is critical for infection include *Bordetella pertussis*, *Haemophilus ducreyi*, *Burkholderia pseudomallei*, *Mycobacterium tuberculosis*, *P. aeruginosa*, *Vibrio vulnificus* and *Yersinia pestis*[2–5]. TadSS genes are widely distributed amongst Gram-negative and Gram-positive phyla[2,6] indicating a broad adaptive advantage for bacteria that acquire Tad pili. Functional diversification includes cell adhesion and colonisation[5,7,8], natural competence[9], twitching motility[10] and recently a key role in contact-dependent prey killing in *Myxococcus xanthus*[11]. The TadSS forms part of the type 4 filament superfamily[12] whose bacterial and archaeal lineage can be traced back to the last universal common ancestor (LUCA). Evolutionary studies show that the TadSS has an Archaeal origin with the closest shared ancestor an EppA-dependent pilus system[13] before interkingdom transfer into bacteria[12].

The TadSS in Gram-negative bacteria typically constitutes 12–14 proteins. The prepilin peptidase TadV matures the Flp1 pilin as well as the TadE and TadF pilins[14]; whilst TadZ has been implicated in targeting the TadSS to the cell poles based on its homology to MinD[15]. Upon assembly, the TadSS is energised by TadA, a cytoplasmic ATPase homologous to GspE in the T2SS and PilB in the type 4 pilus system[16]. Based on similarity to other type 4 filament family proteins, TadA may engage the inner membrane proteins TadB and TadC to couple ATP hydrolysis with the mechanical spooling of inner membrane pilins into the assembling pilus. Whilst Flp1 constitutes the major component of the TadSS pilus[17], the pilins TadE and TadF may also be integrated into the pilus although their precise role is unclear[9,18]. In Gram-negative bacteria, the pilus is expected to pass through the outer membrane via the secretin RcpA, which forms a monophyletic clade[19] distinct from those found in the type II and III secretion systems (T2SS[20], T3SS[21]), the type 4 pilus system[16], and filamentous bacteriophage[22]. Intriguingly, the closest shared common ancestor to RcpA is the secretin PilQ from the type 4 pilus system[23], from where it was presumably repurposed to facilitate passage of the TadSS pilus across the outer membrane of Gram-negative bacteria. In general, secretins[22,24–32] play a critical role in the translocation of macromolecules including toxins, DNA, and pilus filaments across the cell envelope into the external environment. All the secretin types have a closely conserved core fold where β-sheets form an inner and outer barrel. Based on functional specialisation, this

[1]Department of Infectious Disease, Imperial College, London SW7 2AZ, UK. [2]Department of Life Sciences, Imperial College, London SW7 2AZ, UK. [3]Present address: Human Technopole, Milan, Italy. [4]Present address: Singapore Centre for Environmental Life Sciences Engineering, Nanyang Technological University, Singapore, Singapore. ✉e-mail: h.low@imperial.ac.uk

core fold is adapted with modifications typically observed within Gates 1 and 2 that control substrate passage, the inner lip that inserts with variable depth into the outer membrane, a cap gate protruding through the outer membrane[20], and the N-terminus where major modifications are made to periplasmic domains that often form stacked rings. At the C-terminus, some secretins have an S-domain that binds lipoprotein chaperones termed pilotins[33].

In *A. actinomycetemcomitans*, RcpA requires TadD for its expression and assembly into a secretin, and correct insertion into the outer membrane[19]. As TadD is also a predicted lipoprotein it may function as the cognate pilotin for RcpA although this has not been conclusively shown. By sequence, TadD has only one identified homologue (<20% sequence identity) in the type IV filament family - the mannose-sensitive hemagglutinin pilus component MshN[12], which is a predicted tetratricopeptide repeat protein like the type 4 pilus system pilotin PilF[34]. More broadly, neither pilotin architecture nor the mechanism by which pilotins bind their target secretin is conserved[35]. Similarly, their functions vary with some required for outer membrane secretin localisation via the LOL pathway whilst others are also needed for secretin assembly[33,36]. In cases such as the *E. coli* T3SS, inner membrane components function as a platform from which to orchestrate secretin assembly[37,38].

Despite the prevalence of the TadSS in many bacteria and its role in human and animal pathogenesis, it remains relatively understudied with no structural studies undertaken on the secretion apparatus including the outer membrane components. For RcpA, little is known about its architecture and stoichiometry, how it assembles to form the secretin and is targeted to the outer membrane, and which TadSS components it may bind. Whether TadD functions as the RcpA pilotin through direct interaction with RcpA and the nature of the binding site has also not been shown. Here we use cryogenic electron microscopy (cryo-EM) to reveal the architecture of the RcpA secretin in complex with TadD from *P. aeruginosa*. The complex assembles with a mix of C13 and C14 symmetries when either heterologously expressed or purified at native levels from *P. aeruginosa*. We establish the C-terminal tip of RcpA to be crucial for binding TadD within its central groove. Based on current understanding[39], this mode of binding is different to how the type 4 pilus system secretin PilQ binds its cognate pilotin PilF despite a close common ancestry. Light microscopy and biochemistry are used to show that assembly and cellular localisation of RcpA depends on TadD. Collectively, our results provide a molecular mechanism for how TadD facilitates RcpA assembly into secretin and positions it within the outer membrane.

## Results

### RcpA requires TadD for secretin assembly and cell localisation

To test whether *P. aeruginosa* TadD binds directly to RcpA and to probe its role as a putative pilotin that promotes RcpA secretin formation, C-terminally strep-tagged TadD (termed TadD$_{strep}$) and RcpA (termed RcpA$_{strep}$) were heterologously expressed in *E. coli* and purified individually (Supplementary Fig. 1). By SDS-PAGE analysis, TadD$_{strep}$ (27 kDa) and RcpA$_{strep}$ (43 kDa) migrated at their calculated molecular weights whilst forming poorly ordered helical filaments or aggregates, respectively, when visualised by negative stain (NS) EM (Supplementary Fig. 1c). In contrast, when TadD$_{strep}$ and RcpA were co-expressed and purified, SDS-PAGE revealed bands consistent with RcpA-TadD$_{strep}$ complex. For the majority of RcpA, migration through the gel was impeded due to the formation of an SDS-resistant multimer —a behaviour that is characteristic of secretin assembly (Supplementary Fig. 1c). Formation of RcpA-TadD$_{strep}$ complex triggered limited TadD proteolysis with two bands observed on the gel consistent with both a full-length and clipped form (~26 kDa). Abundant particles consistent with secretin channels were also observed by NS-EM (Supplementary Fig. 1c). RcpA secretin formation was therefore dependent on direct association with TadD. To show that RcpA and TadD interact

in a cellular context and to test whether RcpA localisation depends on TadD, fluorescent microscopy studies were undertaken initially in *E. coli*. Constructs for *rcpA-mNeon* and *tadD-mScarlet$_{strep}$* were generated and their capacity to assemble into a secretin complex was verified by purification and EM analysis (Supplementary Fig. 2a). RcpA-mNeon and TadD-mScarlet$_{strep}$ were then expressed either individually or in combination and visualised (Fig. 1a and Supplementary Fig. 2b). RcpA-mNeon formed well-defined punctate foci at the cell poles whilst TadD-mScarlet$_{strep}$ formed cell envelope diffuse patches that sometimes included the cell poles but were not restricted to them. Strikingly, when RcpA-mNeon and TadD-mScarlet$_{strep}$ were co-expressed, RcpA-mNeon now closely tracked the localisation pattern of TadD-mScarlet$_{strep}$. Subsequently, we aimed to verify that RcpA and TadD co-localise in *P. aeruginosa* whilst also using their cellular position as a possible marker for assembled TadSSs in the cell. Native *rcpA* and *tadD* were replaced by *rcpA-mNeon* and *tadD-mScarlet* within the chromosome. The expression of full-length RcpA-mNeon and TadD-mScarlet in *P. aeruginosa* was verified by Western blot (Supplementary Fig. 3a). RcpA-mNeon and TadD-mScarlet formed 0.82 and 1.12 punctate foci per cell, respectively (Supplementary Fig. 3b). Foci localised to the cell envelope with the cell poles generally excluded. In 41% of all fluorescent cells, one or two RcpA-mNeon and TadD-mScarlet foci co-localised (average Pearson's Coefficient r = 0.78) indicative of complex formation (Fig. 1b and Table S4). The number of co-localised foci per cell may also represent how many assembled TadSSs are present in each cell although it cannot be excluded that each co-localised focus comprises multiple TadSSs clustered so that the number per cell may be higher. Overall, given the necessity of RcpA to associate with TadD for both secretin assembly and cellular positioning, our results were consistent with a pilotin role for TadD.

### Cryo-EM structure determination of RcpA secretin in complex with TadD pilotin

To probe the assembly mechanism of RcpA$_{strep}$ secretin in complex with C-terminally FLAG-tagged TadD (termed TadD$_{FLAG}$), single particle cryo-EM was undertaken. RcpA$_{strep}$-TadD$_{FLAG}$ complex were purified by double pulldown affinity chromatography (Fig. 2a) and the sample quality checked by NS-EM. Sample was initially applied to holey carbon grids and vitrified but this yielded secretin particles stripped of TadD$_{FLAG}$ indicating a relatively weak interaction. RcpA$_{strep}$-TadD$_{FLAG}$ complex was therefore applied to a graphene oxide support, which yielded secretin particles that maintained TadD decoration after vitrification (Fig. 2a). Data collection and subsequent processing steps (Table S1 and Supplementary Fig. 4) showed that most particles fitted a C13 symmetry model whilst a subset (~5%) exhibited C14 symmetry. Working with the C13 symmetry bin, a map at 2.7 Å overall resolution was obtained (Fig. 2b, c, and Supplementary Fig. 5). Map quality was sufficient to build a structure for the RcpA secretin between amino acids 131–383 excluding disordered residues between the tip of Gate 1 amino acids 242–255 and the S-domain between amino acids 381–416 (Fig. 2d and Supplementary Fig. 6). For TadD$_{FLAG}$, map quality was insufficient for model building. Closer inspection of the RcpA$_{strep}$-TadD$_{FLAG}$ complex class averages indicated that TadD$_{FLAG}$ became increasingly disassembled from the secretin when the particle was angled towards side view orientations on the graphene oxide support. A second 30°–40° tilted dataset was therefore collected and merged with the original dataset. This combined dataset was symmetry expanded and 3D classified[40]. A focussed refinement using a mask that enveloped three RcpA subunits and three TadD subunits yielded a TadD map (termed TadD$_{FRmap}$) at ~4 Å resolution (Supplementary Fig. 5b) which enabled a TadD structure to be built between amino acids 29–212 (Fig. 2e–g and Supplementary Figs. 7, 8). The resulting RcpA$_{strep}$-TadD$_{FLAG}$ complex structure was 188 Å in diameter with RcpA forming a classical secretin-like channel decorated by TadD subunits around the outside face (Fig. 3a). The RcpA channel was

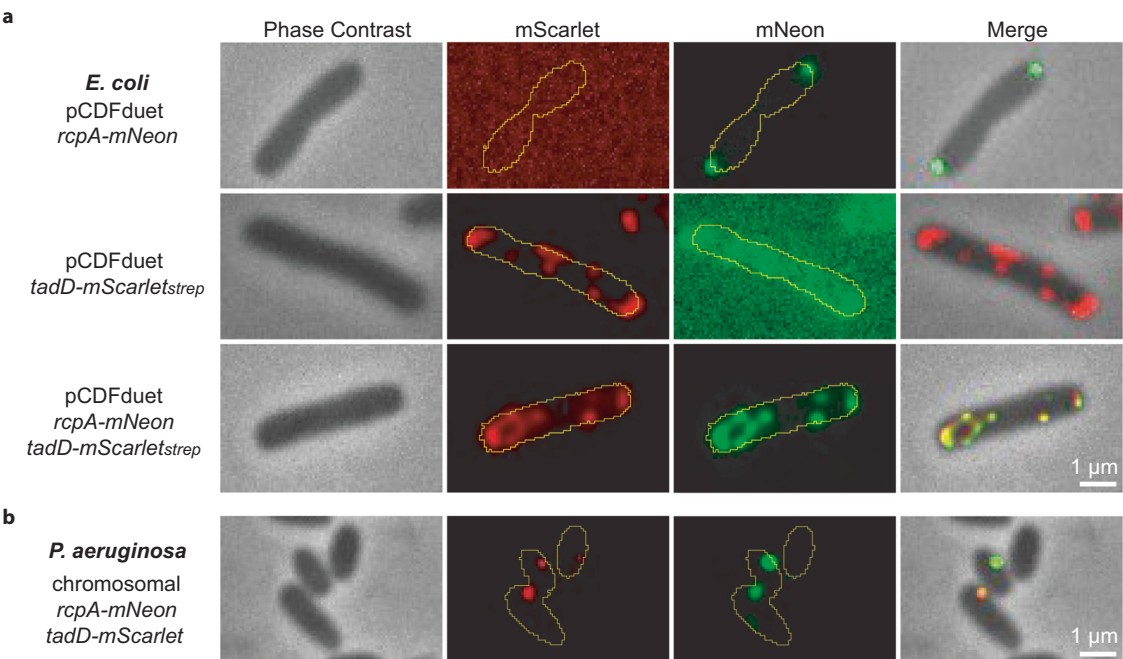

**Fig. 1 | RcpA requires TadD for secretin assembly and cell localisation. a** RcpA-mNeon cellular localisation depends on TadD-mScarlet_strep when heterologously expressed in *E. coli*. In the absence of TadD-mScarlet_strep, RcpA-mNeon accumulates as punctate foci at the cell poles. **b** Under native expression levels, RcpA-mNeon and TadD-mScarlet form punctate foci localised to the cell envelope in *P. aeruginosa*. Three independent replicates were undertaken with similar results.

partially occluded by a central gate with a 31 Å diameter opening formed between the tips of Gate 2 (Fig. 3b). Above the central gate, the inner lip constricted the channel yielding a 62 Å diameter lumen. Each RcpA subunit packed at an angle (Fig. 3a) so that it spans ~80° around the channel circumference (Fig. 3c). Consequently, a single RcpA subunit shared minor packing interfaces with three neighbouring TadD subunits (Fig. 3b, c).

## RcpA secretin in complex with TadD form C13 and C14 symmetries in *P. aeruginosa*

Given a subset of RcpA_strep-TadD_FLAG particles exhibited C14 symmetry (Fig. 4a), we questioned the physiological stoichiometry of this complex in *P. aeruginosa*. We therefore aimed to isolate RcpA-TadD complex directly from *P. aeruginosa* working with native expression levels from the TadSS operon[2]. Accordingly, *tadD* was replaced by *tadD_strep* within the *P. aeruginosa* chromosome. Affinity chromatography was undertaken with the presence of TadD_strep in the StrepTrap eluate verified by Western blot (Fig. 4b). NS-EM analysis of the eluate revealed particles (Fig. 4b) consistent with secretin end views. A NS dataset was collected and processed yielding class averages that intriguingly showed two symmetries with C13 and C14 observed in 2.7:1 ratio (Fig. 4b). Given the high stability of the assembled RcpA secretin as indicated by its resistance to SDS and heat (95°) denaturation (Supplementary Fig. 1c), it is likely that RcpA/TadD secretin complex exists in both symmetries within the *P. aeruginosa* cell with the C13 form the most prevalent. To verify that the strep tag located at the C-terminus of RcpA does not influence secretin stoichiometry, the strep tag was moved to the N-terminus (termed _strepRcpA). Subsequent heterologous expression in *E. coli* and purification of the _strepRcpA/TadD complex yielded class averages (Supplementary Fig. 9a) equivalent to those purified from *P. aeruginosa*. The C14 symmetry bin also comprised a fraction of the particles (7%) which was consistent with purifications using C-terminal strep-tagged RcpA (Fig. 2a). It was therefore concluded that the strep tag location does not affect secretin stoichiometry. In our cryo-EM datasets no side views of the C14 symmetry particles were obtained due to orientation bias on the graphene oxide

support so a 3D reconstruction was not undertaken. However, to model the C14 symmetry particle, end view class averages (0–25°) were back-projected to generate an anisotropic map (Supplementary Fig. 9b), which was used for the rigid body fitting of fourteen RcpA subunits to generate a secretin model. TadD subunits were then positioned in the model in equivalent positions to TadD in the RcpA-TadD asymmetric unit from the RcpA_strep-TadD_FLAG C13 symmetry structure. This resulted in similar packing interfaces between neighbouring TadD subunits as observed in the C13 symmetry structure but importantly with no inter-subunit clashes. The resulting model (Fig. 4c and Supplementary Fig. 9c) superposed closely with a typical C14 symmetry class average (Fig. 4d). Our data therefore supports the assembly of C14 symmetry particles using the same RcpA/TadD asymmetric unit as observed in the C13 model with only local packing adjustments required between neighbouring subunits. Overall, our data is consistent with C13 and C14 symmetries representing the functional stoichiometries for the *P. aeruginosa* TadSS secretin/pilotin complex.

## RcpA has a classical secretin-like fold and a VirB9-like N-domain

The RcpA monomer (Fig. 5a) shares a similar fold with other secretins including those from the type 4 pilus system[25,26], T2SS[27–31,41] and T3SS[32]. Specifically, it has a root-mean-square-deviation (RMSD) Cα = 4.3 Å when aligned against the secretin domain of *P. aeruginosa* type 4 pilus system PilQ[25] (Fig. 5b). The fold includes parallel inner and outer barrels comprised of four β-sheet motifs that interconnect at the secretin base. Extending from the inner barrel, Gates 1 and 2 protrude into the central lumen. Gate 1 is flexible presumably to facilitate hinging between a closed conformation where the channel is largely occluded and an open conformation that facilitates passage of the pilus as for the T3SS[42]. The N-terminus of RcpA constitutes an N-domain (amino acids 28–112) connected to the secretin domain by a 20 amino acid linker (Supplementary Fig. 1a). In other RcpA homologues analysed (Supplementary Fig. 1a and 6) this linker is predicted to be helical and is separated from the secretin base by inclusion of an N3 domain as found in multiple secretin types

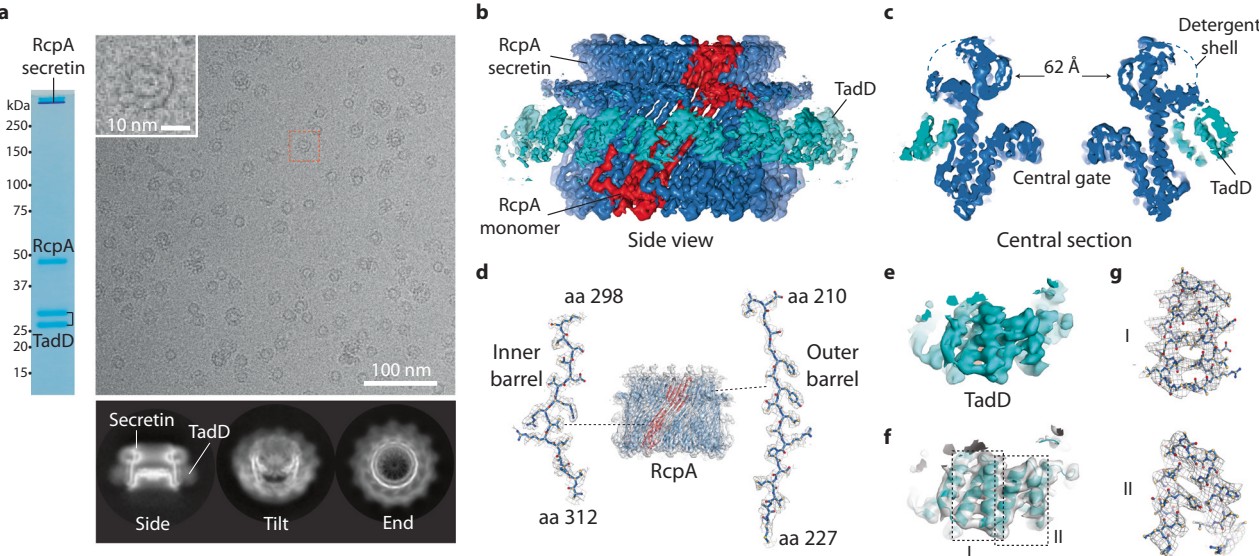

**Fig. 2 | Cryo-EM map and model of RcpAstrep-TadDFLAG complex with C13 symmetry. a** Purification and visualisation of RcpA-TadD complex. (Left) SDS-PAGE analysis of RcpA$_{strep}$-TadD$_{FLAG}$ purified by double pulldown affinity chromatography reveal RcpA$_{strep}$ in both monomeric (43 kDa with C-terminal strep tag) and an SDS-resistant form consistent with secretin assembly. TadD$_{FLAG}$ migrated as two bands consistent with full-length (27 kDa with C-terminal FLAG tag) and a clipped form (~26 kDa). (Right) Typical cryo-EM micrograph showing RcpA$_{strep}$-TadD$_{FLAG}$ particles with associated class averages beneath. The purification was repeated at least three times independently with similar particles obtained.

**b, c** Side and central section view of the RcpA$_{strep}$-TadD$_{FLAG}$ complex cryo-EM map at overall 2.7 Å resolution. No map was resolved for the RcpA N-domain. Map contoured at 6σ. **d** Fit of the secretin model within the RcpA$_{strep}$-TadD$_{FLAG}$ complex map with selected regions zoomed to show side chain detail and build. Map contoured at 5σ. **e** TadD map obtained with a symmetry expansion and focussed refinement strategy (termed TadD$_{FRmap}$). Map contoured at 6.5σ. **f** Fit of the TadD model within TadD$_{FRmap}$. **g** Zoomed regions of TadD$_{FRmap}$ as boxed in **f** showing side chain detail and build.

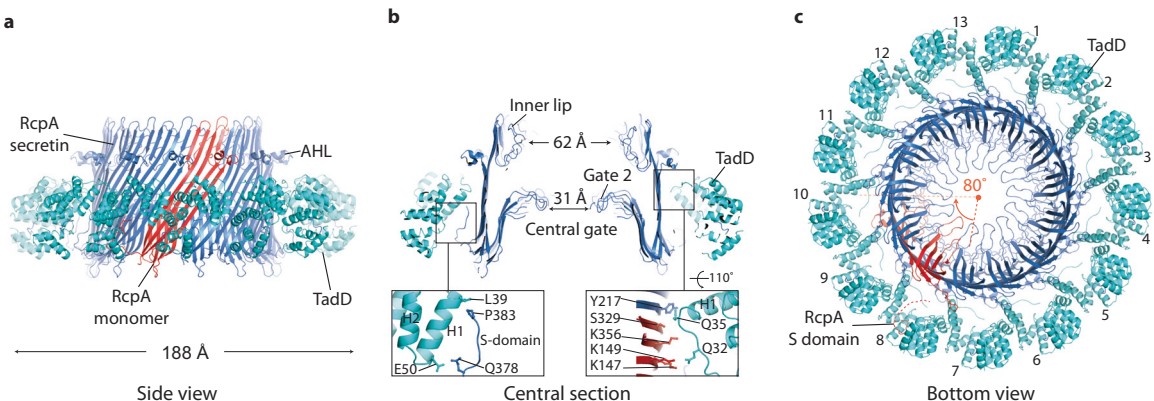

**Fig. 3 | Cryo-EM structure of RcpA$_{strep}$-TadD$_{FLAG}$ complex with C13 symmetry. a–c** Side view, central section and bottom view of RcpA$_{strep}$-TadD$_{FLAG}$ complex. No structure was built for the RcpA N-domain. In **b** zoom panels highlight contact zones between RcpA and TadD. (Left) RcpA S-domain between Q378-P383 runs parallel to TadD H1. (Right) TadD Q35 located at the N-terminus of H1 contacts two neighbouring RcpA subunits coloured in red and blue. **c** One RcpA subunit highlighted in red contacts three neighbouring TadD molecules numbered 8, 9, and 10. AHL amphipathic helical domain.

including those of the T2SS (Fig. 5c, d)[20]. In *P. aeruginosa* RcpA, neither the N-domain nor the linker were resolved in the 2D class averages or map (Fig. 2a–c) indicating high flexibility. This is despite the known presence of the N-domain within the sample as peptides relating to this domain were identified by mass spectrometry. N-terminal domain flexibility is common in secretins with the N0 domains of type 4 pilus system[25] and T2SS[27–30,41] secretins not resolved unless stabilised through the binding of other secretion system components at the secretin base[31,32]. In the absence of an experimentally determined model for the RcpA N-domain, a bioinformatics analysis was undertaken searching for homologues. Interestingly, hidden Markov models prediction[43] indicated close homology (96.2% probability) with the N-terminal domain of the *Xanthomonas citri* type IV secretion system (T4SS) component

VirB9[44] despite having <20% sequence homology. An Alphafold[45] model of the RcpA N-domain (termed RcpA N-domain$_{Alphafold}$) was subsequently generated revealing a barrel-like structure comprising seven β-sheets. RcpA N-domain$_{Alphafold}$ was then used as a search model for homologous structures in the PDB database[46] and was found to share a highly similar fold to *X. citri* VirB9 N-terminal domain with a calculated RMSD Cα = 2.3 Å. *X. citri* VirB9 N-terminal domain forms a periplasmic C14 ring (I-layer) positioned beneath the outer membrane channel components of the T4SS core complex (O-layer). The functional similarity between the T4SS core complex and the TadSS secretin combined with their common positioning within the outer membrane suggests that the RcpA N-domain may form a similar periplasmic ring upon stabilisation with other TadSS components.

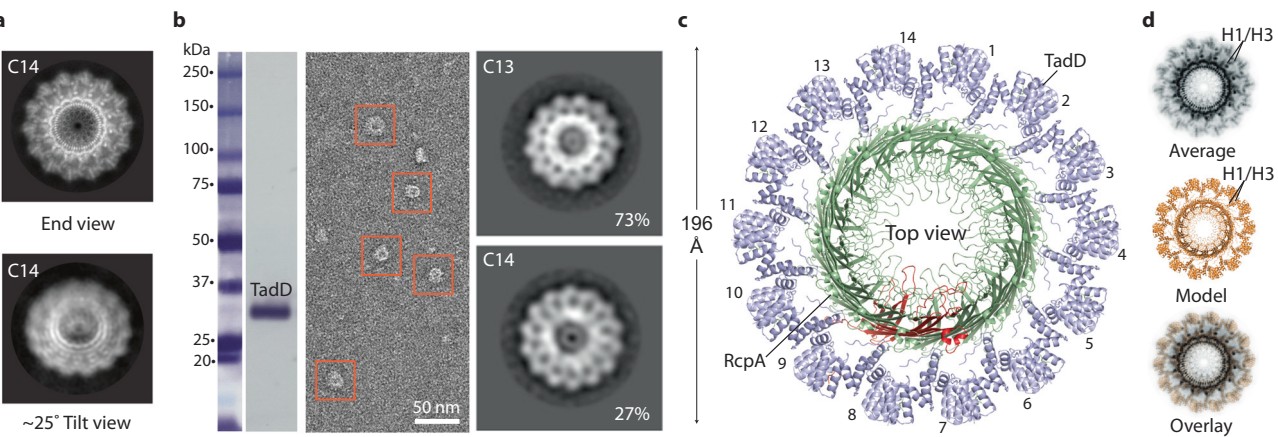

**Fig. 4 | Modelled RcpA-TadD complex with C14 symmetry. a** Cryo-EM 2D class averages of RcpA$_{strep}$-TadD$_{FLAG}$ complex exhibiting C14 symmetry. Samples were purified after heterologous expression in *E. coli*. Orientation bias yielded class averages between 0 and -25˚ tilt. **b** RcpA-TadD$_{strep}$ complex purified directly from *P. aeruginosa* working with native expression levels. (Left) Western blot with anti-strep antibody confirmed the presence of TadD$_{strep}$ in the eluate after purification by affinity chromatography. (Middle) Negative stain EM image of purified RcpA-TadD$_{strep}$ complex. (Right) Class averages showing the presence of both C13 (73%) and C14 (27%) symmetries. The purification was repeated twice independently with similar particles obtained. **c** *P. aeruginosa* RcpA-TadD$_{strep}$ C14 model. **d** Overlay of C14 symmetry model with typical C14 symmetry 2D class average.

## Structure of TadD reveals tetratricopeptide repeats with a PilF-like fold

The structure of TadD constituted 11 helices with helices H1-10 comprising five consecutive tetratricopeptide (TRP) repeats[47]. Collectively, these helices create a crescent shape so that the concave face forms a central groove (Fig. 6a–c). The TadD N-terminus incorporates a signal peptide that is cleaved after amino acid 16 with Cys17 predicted to be lipidated and anchored in the outer membrane (Supplementary Fig. 1b). No map is observed between amino acids 17–27 indicating flexible coil. The model initiates at Glu28 with neighbouring subunit Cys29 in sufficient proximity to Cys97 to form a disulphide bond that stabilises Loop 1 (Fig. 6d) before the start of the TRP repeats. Although *P. aeruginosa* TadD shares just 20% sequence identity with the type 4 pilus system pilotin PilF, hidden Markov models prediction[43] indicated close homology (99.9% probability) with PilF. Analysis of the PilF crystal structure from *P. aeruginosa* showed a similar core fold to TadD although it comprises 13 helices forming six TRP repeats (Fig. 6e). Superposition of TadD with PilF using the common helices H1-10 revealed their close structural relationship with a calculated RMSD Cα = 3.1 Å (Fig. 6f). TadD packs around the secretin periphery with neighbouring subunits forming a circular filament where helix H2 in one subunit is positioned against helix H8 and the loop connecting helices H6 and H7 in the neighbouring subunit. The interface is non-polar mediated by conserved residues A55 and A58 contacting P131 and L163, respectively (Fig. 6b–d).

## The C-terminus of RcpA residues 412–416 bind TadD

After building the TadD model, a section of the TadD$_{FRmap}$ within the positively charged TadD groove (Fig. 6a and Supplementary Fig. 7) remained unassigned. Hidden Markov models prediction[43] as well as structural homology searches[48] showed that in other close structurally related TPR-containing proteins, the groove often mediated protein-protein contacts[49–52]. It was therefore reasoned that part of the C-terminal S-domain of RcpA, predicted to be unstructured, likely constituted the unassigned map section. As the map resolution was insufficient to unequivocally identify those RcpA amino acids forming the interface, a series of RcpA S-domain truncations was generated and their ability to inhibit secretin formation through loss of TadD recruitment tested (Fig. 7a). Truncation of the C-terminal seventeen residues (RcpA$_{aa1-399}$) as well as the last four residues (RcpA$_{aa1-412}$) both resulted in loss of TadD binding and secretin assembly. To confirm the requirement of the last four residues (RcpA$_{aa413-416}$) for TadD

recruitment in a cellular context, *rcpA$_{aa1-412}$-mNeon* and *tadD-mScarlet$_{strep}$* were heterologously expressed in *E. coli* and visualised. RcpA$_{aa1-412}$-mNeon did not co-localise with TadD-mScarlet$_{strep}$ as was observed when using full-length RcpA-mNeon (Fig. 7b). Instead, its localisation pattern matched RcpA-mNeon when individually expressed. Collectively, these studies showed that RcpA$_{aa413-416}$ were essential residues for TadD recruitment. The terminal five RcpA residues were therefore built into the unassigned TadD$_{FRmap}$ section with the register of the bulkier residues supported by the map. TadD helices H3, H5, H7, H9 and H11 form the central groove and interface with RcpA$_{aa412-416}$ (Fig. 6a). Specifically, TadD L108, L111 and V142 form a hydrophobic patch towards which RcpA L414 is oriented (Fig. 6a and Supplementary Fig. 7). Conserved TadD residue R79 is oriented towards and stabilises RcpA terminal residue D416, whilst TadD D139 is positioned stabilising the main chain peptide N-H group of RcpA S415 (Fig. 6a and Supplementary Fig. 7). Note that D139 is a highly conserved asparagine in other TadD systems (Supplementary Fig. 8). Based on the structure and to further verify the assignment of RcpA$_{aa412-416}$ as the key residues mediating TadD binding, cysteine cross-linking assays were undertaken that aimed to promote the formation of targeted inter-subunit disulphide bonds. Given TadD D139 is in a highly conserved position in the centre of the binding groove within helix H7 (Fig. 6b), D139C was targeted for cross-linking to RcpA residues L414C and S415C with sulphur atoms predicted to be 4.9 Å and 3.5 Å apart in the structure, respectively (Fig. 7c). RcpA L414C and S415C mutants were significantly impeded in TadD binding yielding 9% and 14% of RcpA-TadD particles relative to native levels as shown using NS-EM (Fig. 7d). TadD was observed to form helical filaments similar to when TadD was expressed and purified alone (Supplementary Fig. 1c). There was no evidence of inter-subunit cysteine cross-linking in either control mutant despite the addition of the oxidising agent copper phenanthroline (Fig. 7d). Similarly, TadD D139C impeded RcpA recruitment yielding just 2% RcpA-TadD particles relative to native TadD and a total absence of secretin observed by SDS-PAGE (Fig. 7d). Again, there was no evidence of inter-subunit cysteine cross-linking in the control mutant. In contrast, double cysteine mutants TadD D139C with either RcpA L414C or S415C restored RcpA secretin assembly yielding native levels of particles (Fig. 7d). Crucially, in the presence of copper phenanthroline cross-linked bands migrating at 72 kDa and 70 kDa were observed by SDS-PAGE which matched the calculated size of RcpA coupled to either full-length or clipped TadD. SDS-resistant sample also accumulated in the gel well consistent with increased RcpA secretin assembly. The addition of DTT reversed the

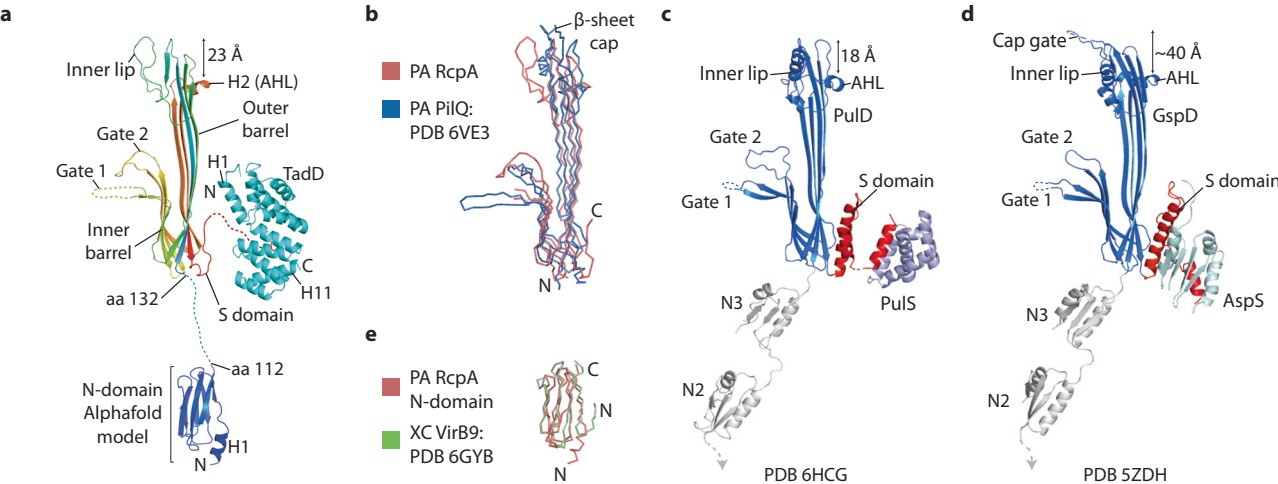

**Fig. 5 | RcpA has a classical secretin-like fold and a T4SS VirB9-like N-domain.** **a** RcpA monomer in complex with TadD pilotin extracted from the C13 symmetry model. AHL - amphipathic helical domain. **b** C-alpha main chain superposition of RcpA and type 4 pilus system PilQ (RMSD 4.3 Å). **c** T2SS *K. pneumoniae* PulD in complex with PulS pilotin. **d** T2SS *E. coli* GspD in complex with AspS pilotin. **e** C-alpha main chain superposition of Alphafold RcpA N-domain model with *X. campestris* T4SS protein VirB9 (RMSD 2.3 Å).

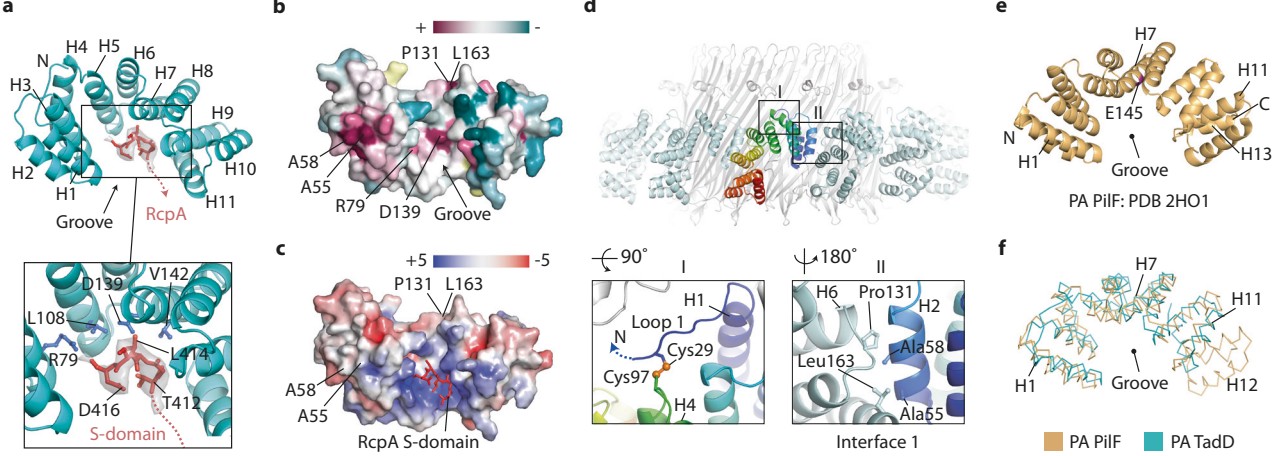

**Fig. 6 | Analysis of TadD and its binding interface with RcpA. a** TadD comprises five tetratricopeptide repeats (H1-H10). Zoom box shows how C-terminal RcpA residues 412–416 with associated TadD$_{FRmap}$ (grey) bind TadD within its central groove. **b** TadD surface rendered to show evolutionary conservation of amino acids. Maroon to green spectrum represents high to variable conservation levels. **c** TadD surface rendered to show electrostatic charge. Blue to red spectrum represent positive to negative charges with units $k_BT/e_c$. **d** Packing arrangement of TadD subunits around the RcpA secretin. Zoom boxes showcase the disulphide bond formation between C29 and C97 which stabilises Loop 1 (left) and the interface between neighbouring TadD subunits (right). **e** Structure of *P. aeruginosa* type 4 pilus system PilF comprising 13 helices. Helix 7 E145 (magenta) located in a similar position to TadD D139 may not mediate PilQ binding[23]. **f** C-alpha main chain superposition of TadD and *P. aeruginosa* PilF with RMSD 3.1 Å calculated using common helices H1-H10.

cross-linking yielding monomeric forms. Collectively, the observation that RcpA$_{aa1–412}$ does not bind TadD and the close association (<3 Å) of TadD D139C with both RcpA L414C and S415C via cysteine cross-linking supports our structure build where RcpA$_{aa412–416}$ constitute the essential TadD binding residues.

## Discussion

Here we use cryo-EM to determine the structure of the TadSS secretin RcpA in complex with TadD from the ESKAPE pathogen *P. aeruginosa*. The study provides a snapshot into TadSS outer membrane architecture (Fig. 8). We show that RcpA/TadD complexes assemble with C13 and C14 stoichiometries both when heterologously expressed in *E. coli* and when purified at native expression levels from *P. aeruginosa*. The structure reveals the RcpA binding interface to be located within the TadD groove. This interface is further dissected using a fusion of biochemistry and cysteine cross-linking assays, which determine the terminal four residues of RcpA to be essential for RcpA-TadD

interaction. This data combined with the observation that TadD is required for RcpA secretin assembly shows that TadD constitutes the RcpA pilotin in the TadSS. Overall, our results provide a molecular framework for understanding how the Tad pilus is channelled through the outer membrane as it undergoes extension and retraction cycling.

RcpA Gate 1 was observed to be flexible with no structure built between P241 and Y256 (Fig. 5a). Similar Gate 1 flexibility was previously observed in T2SS secretins which have 11–15 residues excluded from structures within the Gate 1 tip region (Fig. 5c, d)[27–31,41]. This contrasts with known structures of the type 4 pilus system where rigidity in the Gate 1 tip facilitated model building so that secretins were occluded except for a central ~20 Å oculus[24,25]. However, for the TadSS, the T2SS and the type 4 pilus system, the gating mechanism may work like the T3SS from *Salmonella enterica* where Gate 1 and Gate 2 have been shown to hinge upwards into an open channel conformation[42]. In the case of the TadSS, this conformational change would facilitate passage of the Flp1 pilus.

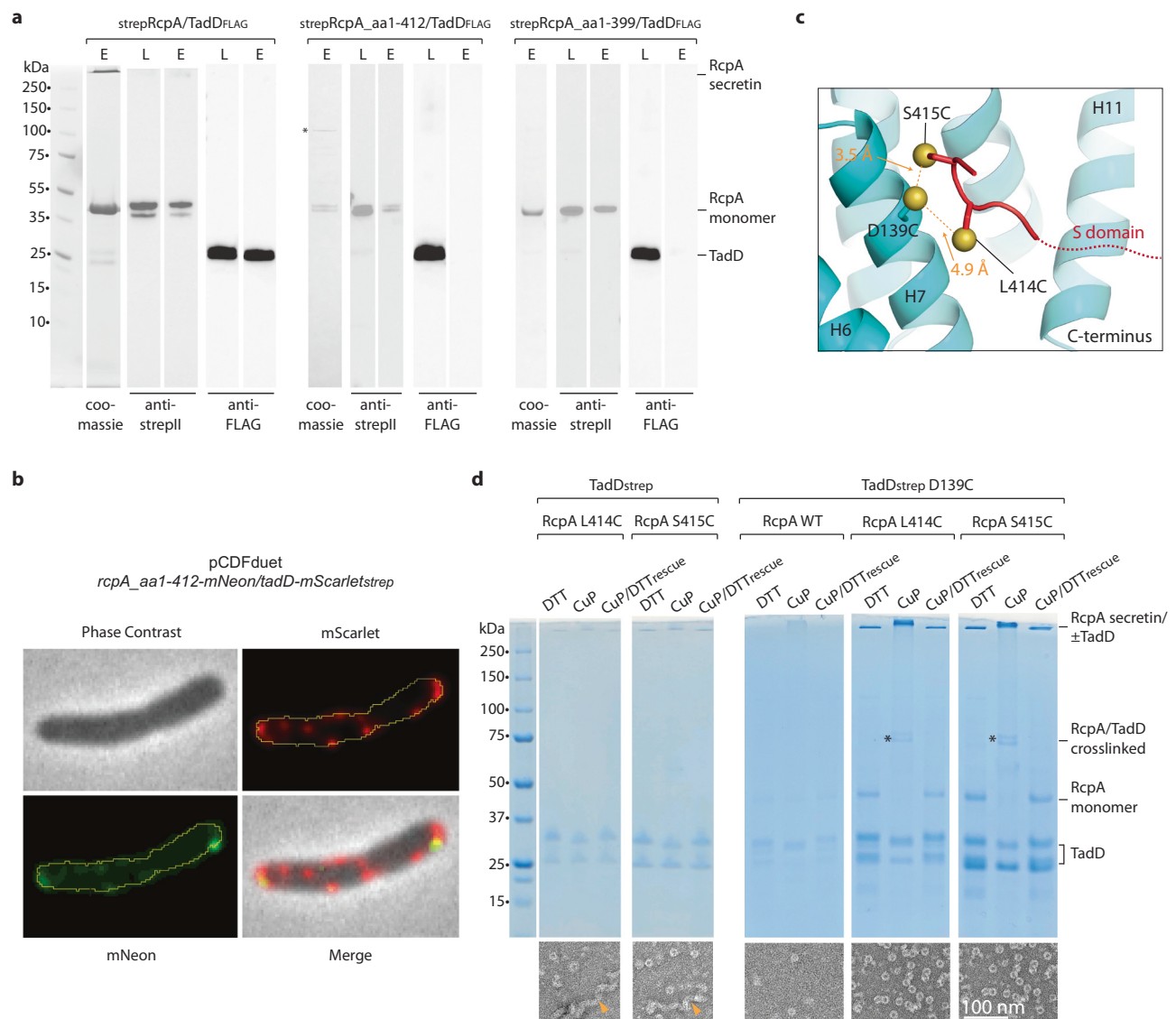

**Fig. 7 | RcpA C-terminal residues 412–416 bind TadD. a** Truncation of the RcpA C-terminus impedes TadD binding and consequently RcpA secretin assembly as shown by SDS-PAGE and Western blot. For each condition a single strep tag pull-down was undertaken with the tag located at the RcpA N-terminus. Asterisk indicates non-specific binding. L = whole cell lysate, E = elution. The experiment was repeated three times independently. **b** RcpA_aa1–412-mNeon does not co-localise with TadD-mScarlet_strep when heterologously expressed in *E. coli* as was observed with full-length RcpA-mNeon. **c** Cartoon schematic showing predicted bond distances between TadD D139C and RcpA S415C or L414C targeted for inter-subunit cysteine cross-linking. **d** Cysteine cross-linking assay. Single cysteine mutants TadD D139C and RcpA S415C or L414C show no cross-linking and almost total loss of RcpA-TadD complex and secretin formation. Representative NS-EM images for these conditions indicate RcpA-TadD complex to be ~2%, 10%, and 10% of native levels, respectively. Orange arrows indicate TadD helical filament as described in Supplementary Fig. 1. Double cysteine mutants TadD D139C with RcpA S415C or L414C show disulphide bond formation (asterisks) and native levels of RcpA-TadD complex and secretin formation as shown by NS-EM. The experiment was repeated three times independently.

The RcpA inner lip is formed from a random coil-turn-coil motif which is different to other known type 4 filament family secretins where the inner lip incorporates helical motifs[24,25,27–31,41]. The RcpA inner lip is flush with the top of the outer barrel β-sheets (Fig. 5a). Based on the position of the conserved amphipathic helical domain (AHL) which is positioned proximal to the inner leaflet (Fig. 5a)[53], both the RcpA inner lip and outer barrel will extend ~23 Å into the membrane thereby spanning the inner leaflet with only minor protuberance expected into the outer leaflet. The rim architecture of the RcpA secretin therefore most closely matches the *Klebsiella*-like family of T2SS secretins (Fig. 5c) rather than the *Vibrio*-like family (Fig. 5d) whose cap gate extends from the inner lip and is sufficient to span at least most of the outer leaflet[20]. Given the close evolutionary heritage between RcpA and PilQ[23], it might follow that these secretins share similar rim architectures. However, whilst PilQ from *P. aeruginosa* does

not have a cap gate, its inner lip is capped by a pair of β-sheets that orient outwards over the top of the outer barrel (Fig. 5b)[25].

Secretins typically have extensive N-terminal domains that protrude into the periplasm[22,24,35,53]. Using bioinformatics and Alphafold[45] analysis, *P. aeruginosa* RcpA was predicted to have a single periplasmic domain which we termed the N-domain. This contrasts with most other type 4 filament family systems such as the T2SS and type 4 pilus system which have a classical N0 domain at the secretin N-terminus (excluding the flexible AMIN domains observed in type 4 pilus systems). In our study, the RcpA N-domain was not observed in 2D class averages or the cryo-EM map despite mass spectrometry indicating its presence in the purified sample. Intriguingly, the N-domain has close homology to the T4SS N-terminal domain of VirB9 as indicated both by primary sequence Hidden Markov models analysis[43] and tertiary structure searches against the PDB database[48]. The VirB9 N-terminal

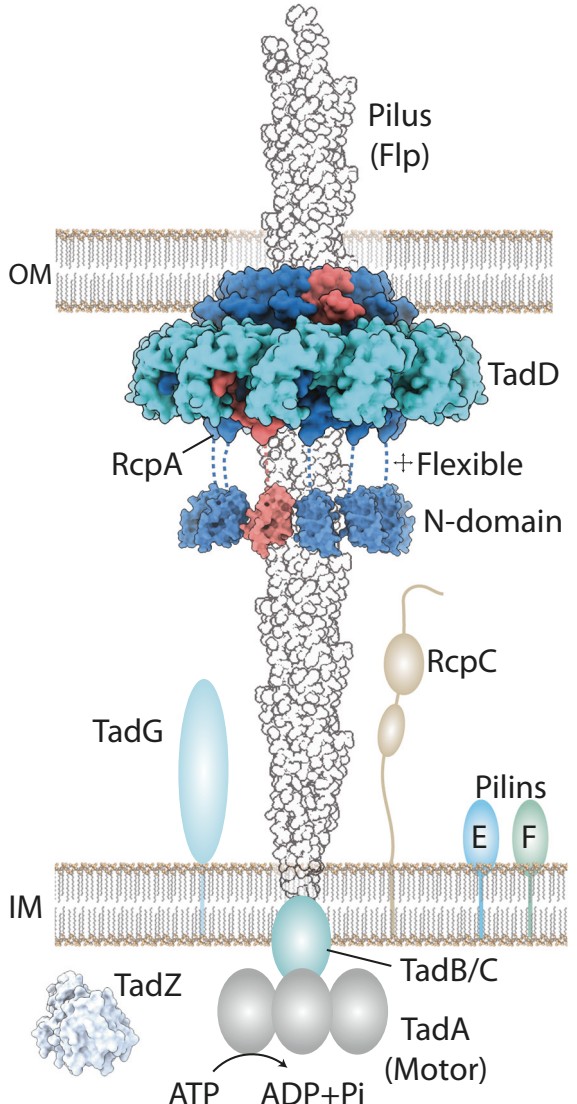

**Fig. 8 | Schematic showing positioning of RcpA-TadD complex within the outer membrane and relative to other TadSS components.** The RcpA N-domain is an Alphafold model and its orientation is modelled based on the *X. citri* VirB9 N-terminal domain periplasmic ring[44]. Hidden Markov model analysis of Flp1 revealed close homology with PilA-N from the *Geobacter sulfurreducens* type 4 pilus filament[79] (as well as archaeal flagellin). The Flp pilus shown therefore represents the ~5 Å diameter helical filament formed by PilA-N subunit from PDB 7TGG. RcpB which is predicted to locate within the outer membrane has been omitted for clarity as has the prepilin peptidase TadV. The MinD-like TadZ was generated from PDB code 3FKQ. RcpC has been shown to localise to both inner and outer membrane fractions[19].

domain forms a ring in the T4SS core complex[44,54], which supports a model where the RcpA N-domain may assemble a similar structure if suitably stabilised by additional TadSS components. In the T2SS, the N-terminal N0 domain of GspD is disordered until bound by the GspC HR domain, which then promotes N0 domain assembly into a ring[31]. As the N-terminus of GspC is tethered within the inner membrane, the GspC-GspD N0 domain interaction provides a mechanism for aligning inner membrane components with the secretin channel. A similar role is likely carried out by PilP in the type 4 pilus system[55,56]. It is therefore possible that an equivalent mechanism will be required in the TadSS to ensure efficient threading of the RcpA secretin by the Flp1 pilus. However, there is no homologue for either GspC or PilP within the TadSS although RcpC has a predicted flexibility and structure that may

be compatible with such a role. Any interaction of TadSS components with the RcpA N-domain may be weak or transient however as no inner membrane complex components were readily pulled down when RcpA/TadD$_{strep}$ complex was purified directly from *P. aeruginosa* (Fig. 4b).

As a broadly conserved theme, secretins are dependent on pilotins for effective localisation to the outer membrane and function. However, neither pilotins nor their mode of binding to their cognate secretin are closely conserved with bespoke pairings previously identified[35]. How TadD recruits RcpA and the nature of this interaction was therefore a key question in this study. The RcpA S-domain is shown to comprise random coil in contrast to many other known secretin-pilotin pairings where the S-domain often has a helical component within the binding interface (Fig. 5c, d)[20,30,53]. Given the close evolutionary relationship between RcpA and PilQ, and that TadD has a PilF-like fold, it may be expected that they share broadly similar binding mechanisms. However, this model is currently not supported as mutation of *P. aeruginosa* PilF E145Y, which is surface exposed on H7 in the centre of the groove, does not inhibit PilQ secretin formation as would be expected if PilF binding was inhibited[39]. Comparison of PilF and TadD structures showed that E145 is in an equivalent position to TadD N138, which is one residue upstream from TadD D139 (Fig. 6a, e). As loss of RcpA binding and secretin formation was induced in the TadD D139C mutant, a similar effect would be expected in the PilF E145Y mutant if PilQ was binding exclusively in the PilF central groove (Fig. 6e). Moreover, PilQ has a short C-terminal S-domain comprising just eight residues. Serial truncation of these residues did not inhibit PilQ secretin formation consistent with another part of PilQ mediating PilF binding[39].

The RcpA/TadD complex from *P. aeruginosa* revealed a mix of C13 and C14 symmetries. This is consistent with other secretin systems where multiple stoichiometries have been observed as for T2SS GspD with C15 and C16 symmetries[27], and the type 4 pilus system with PilQ C14 and C15 symmetries[25]. Given we were able to purify RcpA/TadD complex from *P. aeruginosa* in both C13 and C14 symmetries under native expression levels, this suggests both symmetries may be compatible with channelling the Flp1 pilus. The remarkable symmetry mismatch observed in the outer membrane core complex of the T4SS[54,57,58] suggests that a degree of symmetry promiscuity may be acceptable for other secretion systems to function like the TadSS.

In summary, our study provides a molecular mechanism for how the TadSS secretin RcpA assembles and is coupled by its pilotin TadD to the outer membrane. It reveals a secretin-pilotin mode of interaction that is distinct from those previously described in the T2SS, T3SS and possibly type 4 pilus system despite a close evolutionary relationship. It serves as an additional paradigm where TPR-containing proteins, reminiscent of BamD in the BAM complex[59,60], function as key factors in the assembly and stabilisation of outer membrane complexes. Our study focuses on the TadSS from *P. aeruginosa* which has been shown to be essential for cell adherence to human epithelial cells[5]. It provides a basis for further TadSS mechanistic studies such as how inner membrane components might link with the outer membrane RcpA/TadD complex and ultimately how Flp1 pilus cycling occurs.

## Methods
### Cloning and strain generation
Plasmids, primers and bacterial strains used or generated in this study are listed in Tables S2 and S3. RcpA from *P. aeruginosa* PAO1 was cloned into pASK3C (IBA) with and without a C-terminal strep tag (termed pASK3C-RcpA$_{strep}$ or pASK3C-RcpA, respectively) or with a N-terminal strep tag (termed pASK3C-$_{strep}$RcpA). TadD was cloned into pCDF with a C-terminal FLAG tag (termed pCDF-TadD$_{FLAG}$) or C-terminal strep tag (termed pCDF-TadD$_{strep}$). Plasmid mutagenesis and all clones were generated using the Gibson isothermal DNA

assembly protocol[61]. To generate *P. aeruginosa* strains, immobilised *E. coli* CC118λpir[62] and *E. coli* 1047 pRK2013[63] were used to conjugate the integrative suicide vector pKNG101[64] into *P. aeruginosa* PAO1. Transformants were selected on Vogel-Bonnel Medium agar plates in the presence of two mg/mL streptomycin and subsequently counter-selected on LB-agar supplemented with 20% sucrose. Final *P. aeruginosa* PAO1 mutant clones containing *tadD-mScarlet*, *rcpA-mNeon* or *tadD*~strep~ gene fusions at native loci were verified by PCR and sequencing to ensure the desired chromosomal modification.

### Light microscopy colocalization studies in *E. coli*
pCDF-TadD~strep~ was modified to incorporate mScarlet between TadD and the C-terminal strep tag (termed pCDF-TadD-mScarlet~strep~). Due to undesirable background autofluorescence from anhydrotetracyline (AHT) required to induce protein expression in pASK3C vector, RcpA with a C-terminal mNeon fusion was generated in pCDF (termed pCDF-RcpA-mNeon). pCDFduet was then used to host TadD-mScarlet~strep~ in one cloning site and RcpA-mNeon in the second cloning site (termed pCDFduet-TadD-mScarlet~strep~/RcpA-mNeon). *E. coli* AI Oneshot cells were transformed with the desired plasmid and plated on LB-agar supplemented with 50 μg/mL spectinomycin. Fresh colonies were picked and grown at 27 °C with 200 rpm shaking to $OD_{600} = 0.4$ in one litre of M9 media supplemented with 10% 2xYT, 0.01% casamino acids and 50 μg/mL spectinomycin. Protein expression was induced with 0.5 mM isopropyl β-D-1- thiogalactopyranoside (IPTG) and 0.08% arabinose and the cultures were incubated for an additional 60 minutes. 3 μL of bacterial sample was applied onto microscope glass slides mounted with M9 media 1.5% agar pads and visualised. Alternatively, TadD-mScarlet~strep~/RcpA-mNeon complex was purified by affinity chromatography single pulldown according to the protocol outlined below (see RcpA-TadD expression and purification).

### Light microscopy colocalization studies in *P. aeruginosa*
*P. aeruginosa* PAO1 was chromosomally modified to code for RcpA and TadD with mNeon or mScarlet fused to their C-termini, respectively. Strain glycerol stocks (Table S2) were streaked onto LB-agar plates and incubated overnight at 37 °C. One colony was inoculated in R2 media overnight at 37 °C with shaking. Cells were diluted to $OD_{600} = 0.1$ in 25 mL of fresh R2 and grown again for approximately 36 hours at 18 °C with shaking. 3 μL of *P. aeruginosa* PAO1 *rcpA:rcpA.mNeon tadD:tadD.mScarlet* were directly applied onto microscope glass slides mounted with 1.5% R2A pads and visualised. Alternatively, the culture was centrifuged and the pellets analysed by western blot as outlined below. For both *E. coli* AI and *P. aeruginosa* PAO1, imaging was performed using a Zeiss Axio Observer Z1 microscope fitted with an Orca Flash 4 V2 CMOS camera (Hamamatsu) and a Plan-Apo 63/1.4 Ph3 oil objective (Zeiss). Images exposed for 80 milliseconds were acquired using phase-contrast and epifluorescence microscopy (485 nm and 555 nm for mNeon and mScarlet, respectively). Fluorescent channels were aligned with ZEN black (Zeiss) software using as reference sun-resolution beads images. The aligned stacks were corrected for signal bleaching and deconvoluted using Huygens Essential software (Scientific Volume Imaging, Netherlands). Pearson Cross-correlation values were obtained using JACOP plugin in Fiji[65,66]. To obtain co-localisation percentages, three images from three independent experiments were divided into four areas. For each area, an $8 × 12$ μM Region Of Interest (ROI) was randomly selected with cells ($n = 165$) manually counted and co-localisation statistic obtained with JACoP.

### RcpA-TadD expression and purification
For cryo-EM studies, 2xYT medium supplemented with 50 μg/mL spectinomycin and 30 μg/mL chloramphenicol was inoculated with *E. coli* C43 (DE3) *pspA:kan^r^* cotransformed with pCDF-TadD~FLAG~ and pASK3C-RcpA~strep~. This C43 strain had previously been modified to incorporate a *pspA* gene knockout via a Lambda Red recombinase

strategy[67]. The culture was grown at 37 °C with shaking at 200 rpm until $OD_{600} = 0.6$. RcpA~strep~-TadD~FLAG~ complex expression was induced by adding 0.55 mM IPTG and 200 μg/L AHT (IBA) while decreasing the temperature to 17.5 °C for 16 hours. Cells were harvested by centrifugation at $4500 × g$ (rotor JS-4.2, Beckman) for 20 minutes at 4 °C. During purification, the sample was kept at 4 °C unless otherwise specified. Cells were resuspended in cold lysis buffer (50 mM Hepes pH 8, 50 mM NaCl, 5 mM EDTA and two tablets of EDTA-free protease inhibitor, Roche) plus 0.5 mg/mL lysozyme and 0.1 mg/mL DNase I (Sigma) and stirred at room temperature for 20 minutes. Cells were lysed by sonication (SONICS Vibra cell) on ice for 12 minutes with two-second bursts at 70% amplitude. The cell lysate was clarified by centrifugation at $30,910 × g$ (rotor JA-25.50, Beckman) for 20 minutes and the supernatant ultracentrifuged at $167,574 × g$ (rotor type 45Ti, Beckman) to collect the membranes. 40 mL of extraction buffer (50 mM Hepes pH 8, 100 mM NaCl, 1 mM EDTA, 1% w/v DDM) was used to extract the membrane proteins with a one hour incubation at room temperature. Insoluble material was removed by ultracentrifugation at $132,380 × g$ (rotor Type 70.1Ti, Beckman) for 15 minutes. The supernatant was loaded onto a 1 ml StrepTrap (Cytiva) equilibrated with wash buffer (50 mM Hepes pH 8, 100 mM NaCl, 1 mM EDTA, 0.06% w/v DDM). After washing, the sample was eluted with wash buffer supplemented with 2.5 mM desthiobiotin (Sigma). The eluted fractions were pooled and incubated with ANTI-FLAG M2 Affinity Gel (Sigma) for one hour. After washing, the sample was eluted with wash buffer supplemented with 0.1 mg/mL 3xFLAG peptide (Sigma). Sample purity and quality were evaluated by SDS-PAGE and NS-EM. Samples for SDS-PAGE were not heated at 95 °C before loading. Heating had little effect on the oligomeric state of the RcpA secretin, with the complex remaining in the gel well during electrophoresis.

For biochemical and light microscopy studies an equivalent protocol was followed as described above but with only single strep tag pulldowns undertaken. Appropriate antibiotic was added depending on the chosen expression plasmid.

### Native RcpA-TadD~strep~ purification from *P. aeruginosa* PAO1
*P. aeruginosa* PAO1 carrying a chromosomal strep tag inserted at the C-terminus of *tadD* were grown as previously described in nine litres of R2 media. RcpA-TadD~strep~ was purified as described above for *E. coli* purification with some variations. Cells were resuspended in cold lysis buffer and processed through two cycles of an Emulsiflex homogeniser (Avestin, USA) operating at 20,000 lb/in². Membrane proteins were extracted with extraction buffer supplemented with 0.6% DDM and the clarified sample loaded onto 0.6 mL of Strep-Tactin Sepharose resin (IBA). Due to low expression levels, the eluted sample was evaluated by Western blot against TadD~strep~ and analysed by NS-EM.

### *P. aeruginosa* PAO1 sample preparation for western blot
*P. aeruginosa* PAO1 wild-type and mutant strain cell pellets were resuspended and incubated for one hour at room temperature in lysis buffer supplemented with 0.1 mg/ml DNase I and 0.5 mg/ml lysozyme. The cells were lysed through sonication (SONICS Vibra cell) with four consecutive 30 seconds bursts at 40% amplitude, each of which was followed by two minutes incubation in an ice bath. The resulting bacterial lysates were mixed with loading dye and analysed via SDS-PAGE and western blot.

### Western blots
After SDS-PAGE, gels were transferred onto polyvinylidene difluoride (PVDF) membranes using an iBlot 2 Dry Blotting System (Invitrogen). The membranes were blocked for one hour in TBS buffer (200 mM NaCl, 20 mM Tris pH 7.5) supplemented with 0.1% Tween-20 and 4% milk powder (anti-mScarlet and anti-FLAG) or 4% bovine serum albumin (anti-mNeon and anti-Strep). The primary antibodies (anti-mNEON from Chromotek diluted 1:500, anti-mScarlet from Chromotek diluted

1:2000, anti-Strep from IBA diluted 1:1000, and anti-FLAG from Sigma-Aldrich diluted 1:1000) were incubated for two hours at room temperature under gentle agitation. The PVDF membranes were washed three times in TBS-0.01% Tween-20 for 10 minutes. The secondary antibody, IgG conjugated with Alkaline Phosphatase (Sigma) was diluted 1:5000 and incubated for 1 hour at room temperature under gentle agitation. The PVDF membranes were washed again twice in TBS-0.01% Tween-20, and once in TBS buffer. The membranes were developed using BCIP/NBT alkaline phosphatase substrate (Sigma).

### RcpA-TadD cysteine cross-linking assay
Purified RcpA-TadD cysteine mutants were incubated for one hour at room temperature with either 10 mM DTT or 1 mM ortho-Cu(II) 1,10-phenanthroline (CuP, stock 10 mM in 20% ethanol). The samples that underwent CuP cross-linking were subsequently treated with 10 mM DTT for an additional hour at room temperature to rescue the cross-linked complexes. The remaining free cysteines were blocked by the addition of 10 mM N-ethylmaleimide (NEM, stock 0.5 M in 100% ethanol). The samples were evaluated by SDS-PAGE and NS-EM. Particle counting reported in Fig. 7d was undertaken on five micrographs per mutant using Laplacian-of-Gaussian blob detection in Relion 4.0.

### Negative Stain EM sample preparation, data acquisition and processing
4 μL of sample was applied onto glow-discharged carbon-coated 300 mesh copper grids (Agar Scientific) and incubated for 40 seconds. The grids were washed with three drops of distilled water and stained with three drops of 2% uranyl acetate. Images were acquired on a FEI Tecnai TEM 120 kV equipped with a TVIPS 4 K 416XF camera at 2.58 Å pixel size. For RcpA-TadD$_{strep}$ purified from *P. aeruginosa* PAO1 (Fig. 4b), 1142 images were collected. Images were CTF estimated using CTFFIND-4.1[68] and processed with Relion 3.1[69]. A total of 18,094 particles were extracted and sorted during 2D classification resulting in a final stack of 1587 particles displaying both C13 (1160 particles) and C14 (427 particles) symmetries. For $_{strep}$RcpA-TadD (note the strep tag at the RcpA N-terminus here) expressed and purified from *E. coli*, 729 images were collected yielding 842,097 extracted particles. After 2D classification, 49,581 cleaned particles displayed C13 (45,980) and C14 (3601) symmetries (Supplementary Fig. 9A).

### Cryo-EM preparation and data collection
3.5 μL of RcpA$_{strep}$-TadD$_{FLAG}$ complex was incubated for one minute on a graphene oxide-coated R2/2 Quantifoil copper grid (Electron Microscopy Science) prior to vitrification in liquid ethane using a Vitrobot Mark IV (FEI). Data acquisition was performed at 300 kV on a Titan Krios electron microscope (LonCEM, The Frances Crick Institute, London, UK) equipped with a Gatan K3 detector operating in counted super-resolution mode at a calibrated pixel size of 0.55 Å. A total of 5892 micrographs were collected at defocus −0.8 to −2.5 μm with a total dose of 50 e⁻/Å² and an exposure time of 4.2 seconds fractionated between 40 frames. For the tilted collection, RcpA$_{strep}$-TadD$_{FLAG}$ complex was vitrified on carbon-coated R2/2 Quantifoil gold grids. A total of 7323 and 537 micrographs were taken at 30° and 40° stage tilt, respectively, at target defocus −1 to −2.5 μm. A total dose of 50 e⁻/Å² was used and an exposure of 4.4 seconds fractionated between 46 frames.

### Cryo-EM data processing
The workflow is described in Supplementary Fig. 4. The non-tilted RcpA$_{strep}$-TadD$_{FLAG}$ complex vitrified on graphene oxide was processed first (Dataset 1). Movies were aligned with MotionCor2[70], CTF estimated with CTFFIND-4.2 and images processed with Relion 3.1. Particles were picked from a subset of 1000 micrographs using Laplacian-of-Gaussian blob detection to generate 2D class averages,

which were used for reference-based picking on the entire dataset. After six rounds of 2D classification, class averages were obtained from 145,339 particles showing end, tilted and side view orientations. C14 end views were removed. Two rounds of 3D classification in C1 were undertaken resulting in 119,534 particles which refined to 4.0 Å all in C1. This map revealed C13 symmetry and it was used as a starting reference for a subsequent 3D refinement with imposed C13 symmetry. Contributions from the micelle region were removed through particle subtraction. After iterative rounds of CTF refinement and particle polishing, a 2.7 Å resolution map was obtained and used for RcpA model building. For the tilted dataset required to increase TadD occupancy within the RcpA$_{strep}$-TadD$_{FLAG}$ complex (Dataset 2), movie frames acquired at 30° and 40° tilted stages were aligned separately using MotionCor2 and the CTF estimated with cryoSPARC Patch CTF[71]. The CTF values were then imported into Relion 4.0[72]. After 2D classification, 77,316 cleaned particles were merged with the 145,339 particle stack from Dataset 1. The combined stack was 3D classified and refined to 4.0 Å in C1. The stack was symmetry expanded in C13 and a 3D classification without alignment was undertaken using a mask encompassing three RcpA-TadD asymmetric units. The particles from the 3D class showing high-resolution detail were used for a particle subtraction step which removed density contributions outside of the three RcpA-TadD asymmetric units. A local refinement was performed resulting in an overall 3.6 Å resolution map (termed TadD$_{FRmap}$ and used for TadD model building). All 3D refinement jobs utilised SIDESPLITTER[73]. Map local resolution was estimated using ResMap[74] embedded in Relion 3.1 or 4.0.

### Model building and refinement
The RcpA monomer structure was built using Coot[75]. SignalP 5.0[76] predicted RcpA amino acids 1–32 as the signal peptide and no supporting map was observed for amino acids 33–130, which constitute the N-domain and linker. The RcpA model comprised amino acids 131–416 excluding amino acids 242–255 and S-domain amino acids 381–411. Using Phenix[77], a C13 symmetry operator was applied to the monomer to reconstitute the secretin. From here, three adjacent subunits were refined with iSOLDE[78] and the central subunit then multiplied with a C13 symmetric operator. The resulting secretin model was real-space refined in Phenix. For building the TadD structure an initial Alphafold model was generated and three copies were fitted into TadD$_{FRmap}$. Three RcpA subunits were rigid body fitted into the map to facilitate modelling of RcpA-TadD interfaces. Amino acids 412–416 from a fourth RcpA subunit were built into one of the TadD subunits for completion (Supplementary Fig. 7). The TadD structure was modelled using iSOLDE and real-space refined in Phenix. The central RcpA-TadD asymmetric unit was then fitted by superposing the RcpA subunit onto one RcpA subunit within the RcpA C13 secretin structure. Finally, a C13 symmetry operator was applied to the asymmetric unit to generate the final RcpA$_{strep}$-TadD$_{FLAG}$ complex structure.

### Reporting summary
Further information on research design is available in the Nature Portfolio Reporting Summary linked to this article.

## Data availability
The data that support this study are available from the correspondence upon request. Cryo-EM maps produced in this study have been deposited in the Electron Microscopy Data Bank (EMDB) under accession codes EMD-16810 (RcpA secretin build) and EMD-16818 (TadD$_{FRmap}$ used for TadD build). Atomic coordinates have been deposited in the Protein Data Bank (PDB) under accession code 8ODN (C13 symmetry RcpA$_{strep}$-TadD$_{FLAG}$ structure). The source data underlying Figs. 2a, b, 7a and d, and Supplementary Figs. 1, 2a and 3a

are provided as a Source Data file. Source data are provided in this paper.

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

## Acknowledgements

Cryo-EM data was collected at Diamond Light Source and London Consortium for CryoEM (LonCEM). We thank Nora Cronin (LonCEM, The Francis Crick Institute) for cryo-EM data collection support, Paul Simpson for in-house EM support, and Suhail Islam for in-house computational support. We thank Diamond for access and support of the cryo-EM facilities at the UK National Electron bio-imaging centre (eBIC) funded by the Wellcome Trust, MRC and BBSRC. We thank David Gaboriau (Imperial College FILM facility) for light microscopy assistance. This work was funded by a Wellcome Trust Senior Research Fellowship (215553/Z/19/Z) to HL.

## Author contributions

M.T. and H.L. designed experiments. M.T. undertook all experiments and generated the data. M.R. and A.F. supported P. aeruginosa fluorescent strains generation. H.L. supervised the work and wrote the paper with contributions from M.T.

## Competing interests

The authors declare no competing interests
