## [Peer Review File · Nature Communications]

Assembly mechanism of a Tad secretion system secretin-pilotin complexReviewers' Comments:

Reviewer #1:

Remarks to the Author:

In this study, Tassinari and co-workers report the cryo-EM structure of the outer-membrane complex from the *P. Aeruginosa* TAD pilus, an important appendage involved in biofilm formation in this major human pathogen. Specifically, they have obtained the structure of the secretin pore RcpA, bound to its pilotin partner TadD. Using a set of elegant biochemistry and fluorescence microscopy experiments, they show that TadD is necessary for RcpA oligomerization, and identifies the determinants for their interaction. Collectively this is a very strong study, that makes important contributions to our understanding of the Tad plus, and of the structure and assembly of secretin complexes.

There are however a number of elements that require some additional information/clarifications, to facilitate the understanding of this study:

- I find the introduction a bit confusing, jumping from the secretin, to the Tad, back to the secretin. It would gain from being better organized. In addition, the authors should include a more thorough description of our current knowledge of the structure of secretins and pilotins.

- The first section (Line 105-onward) requires a broader description of the known role of pilotins: In many systems, its role is to transport the secretin to the OM, via the LOL pathway, NOT to promote oligomerization (eg. PulS/PulD, see Guilvout et al, EMBO, 2006). In addition, some secretins do not have pilotins (eg. EscC, see Gauthier et al, Infect Immun, 2003). These should be mentioned, and the reported results should be discussed accordingly.

- Along this line, the secretin in most systems forms SDS-resistant oligomers, as does RcpA in this study. However, the oligomer usually dissociates when the sample is boiled. Is this the case for RcpA, are the various gels shown with boiled, or unboiled, samples? This should be discussed.

- The authors show a schematic representation of the *Caulobacter* RcpA orthologue in figure S1a, and some preliminary biochemical analysis in figure S8D. However it is not obvious from the later figure that the complex is a 14-mer as the authors claim, and overall this doesn't contribute very much to the story, so I would recommend removing this altogether.

- Figure 7 is quite confusing overall. I think panel A would really gain from a WB (cell lysate and pull-down), to confirm that both proteins are expressed in lanes 5 and 7, and there isn't some oligomer stuck in the gel well, that isn't seen here. I also think panels 7D and 7E should be inverted/merged, for clarity. The labels of panel 7D (I-V) are not explained, and therefore this is hard to interpret.

- The model building for the region of RcpA present in the TadD groove was obviously very challenging, and the authors go to great lengths to present a suitable model, verified experimentally. However, for clarity panel 6a should be moved to figure 7. A close-up view of the density would really help convince us that the model is credible, and the local resolution should be cited. Did the authors try to generate a model with AlphaFold multimer?

- Figure 8: TadZ should be included (with the previously-determined crystal structure of its cytoplasmic domain). Also, RcpC is usually localized to the OM, not the IM as depicted here. I am aware of the discussions in the community that it may not be the case, but the representation chosen here should be justified (also applies to lines 380-381).

Some additional minor comments:

- I think table S1 should be altered: Currently it looks like there are two refined models, one obtained with each dataset. However, there is only one atomic model deposited, which (if I understand correctly) was obtained by merging the two datasets. This needs to be altered accordingly.
- Figure S3a: The RcpA band appears at the size of a monomer, when co-expressed with TadD. This is contrary to the results reported above, where it forms SDS-resistant oligomers. This needs to be explained. Was this sample boiled, unlike the previous ones?
- In figure S4, the set of particles with C14 symmetry is not show. It would be helpful to show this here.
- Figure S5a: the dotted line for the resolution criterion is missing.
- This is not critical here, but did the authors make a model of the RcpA N-domain 13/14-mer, with AlphaFold multimer or using the VirB structure?

Reviewer #2:

Remarks to the Author:

TadSS and secretin family systems are extremely important virulence factors that allow bacterial pathogens to cause disease, persist, and to share DNA. Therefore, how they are assembled is not only a fundamentally interesting question, but is also important to solve to help progress the development of novel antimicrobials.

Here Tassinari et al conduct well executed experiments to uncover the structure of the outer membrane complex of the *P. aeruginosa* TadSS – RcpA-TadD – to high resolution. The methods are appropriate and it was excellent to see the authors going the extra mile to conduct focused classification to gain extra insight into the TadD structure to aid the modelling. Coupling CryoEM with both light-microscopy and crosslinking techniques also made for a thorough analysis accompanied by the appropriate controls to support the conclusions. Overall, the major novel discoveries are that the channel can exist in two different symmetries and the discovery of the RcpA-TadD binding interface. I only have some minor thoughts and minor points for the authors to consider:

Another TPR containing lipoprotein is BamD and it has been shown that it is required for the assembly of the essential BamA outer membrane barrel protein. Similarly, BamD may bind to beta sheets during the folding of barrel protein substrates. It is interesting to see the potential overlap in function of TadD and BamD in their ability to help assemble RcpA. Perhaps there is an evolutionarily conserved functional overlap?

1) It is not clear what the gel in Fig 4B is (Coomassie? Blot?). Also, why is there no RcpA present?

2) Line 298: "...with sulphur atoms (predicted to be) 4.9A ..."

3) Line 308 to 311 and Fig 7E: There is also a corresponding increase in the SDS resistant fraction of RcpA secretin TadD high molecular weight bands in the paired cys samples when CuP treated that might be worth mentioning. (Seemingly, the disulfide bond stabilises the assembled form which goes along well with the conclusions.)

(4) The Discussion is very long. Maybe consider condensing.

We would like to thank the reviewers very much indeed for taking the time to critique our manuscript so thoroughly and constructively. The manuscript has benefited from your comments.

REVIEWER COMMENTS

Reviewer #1 (Remarks to the Author):

In this study, Tassinari and co-workers report the cryo-EM structure of the outer-membrane complex from the *P. Aeruginosa* TAD pilus, an important appendage involved in biofilm formation in this major human pathogen. Specifically, they have obtained the structure of the secretin pore RcpA, bound to its pilotin partner TadD. Using a set of elegant biochemistry and fluorescence microscopy experiments, they show that TadD is necessary for RcpA oligomerization, and identifies the determinants for their interaction. Collectively this is a very strong study, that makes important contributions to our understanding of the Tad plus, and of the structure and assembly of secretin complexes.

There are however a number of elements that require some additional information/clarifications, to facilitate the understanding of this study:

- I find the introduction a bit confusing, jumping from the secretin, to the Tad, back to the secretin. It would gain from being better organized. In addition, the authors should include a more thorough description of our current knowledge of the structure of secretins and pilotins.

The introduction has been revised so no longer goes secretin->Tad->secretin. Paragraphs have been included describing secretins and pilotins.

- The first section (Line 105-onward) requires a broader description of the known role of pilotins: In many systems, its role is to transport the secretin to the OM, via the LOL pathway, NOT to promote oligomerization (eg. PulS/PulD, see Guilvout et al, EMBO, 2006). In addition, some secretins do not have pilotins (eg. EscC, see Gauthier et al, Infect Immun, 2003). These should be mentioned, and the reported results should be discussed accordingly.

The suggested revision has been included as outlined above.

- Along this line, the secretin in most systems forms SDS-resistant oligomers, as does RcpA in this study. However, the oligomer usually dissociates when the sample is boiled. Is this the case for RcpA, are the various gels shown with boiled, or unboiled, samples? This should be discussed.

Samples for SDS-PAGE were not heated at 95°C before loading. Heating had little effect on the oligomeric state of the RcpA secretin, with the complex remaining in the gel well during electrophoresis. These sentences have now been included in Methods. Heat resistance is also described in the main text L198 and in Supplementary Figure 1C legend.

- The authors show a schematic representation of the *Caulobacter* RcpA orthologue in figure S1a, and some preliminary biochemical analysis in figure S8D. However it is not obvious from the later figure that the complex is a 14-mer as the authors claim, and overall this doesn't contribute very much to the story, so I would recommend removing this altogether.

The *Caulobacter* data has been removed from the manuscript.

- Figure 7 is quite confusing overall. I think panel A would really gain from a WB (cell lysate and pull-down), to confirm that both proteins are expressed in lanes 5 and 7, and there isn't some oligomer

stuck in the gel well, that isn't seen here. I also think panels 7D and 7E should be inverted/merged, for clarity. The labels of panel 7D (I-V) are not explained, and therefore this is hard to interpret.

The experiments relating to Figure 7A have been redone now using N-terminally Strep tagged RcpA and C-terminally Flag tagged TadD constructs. For all conditions, single pulldowns were carried out using strepRcpA. Western blots using anti-StrepII and anti-Flag antibodies confirm the expression of both RcpA and TadD where expected, and show that there is no oligomer stuck in the gel well in the truncated conditions. The negative stain images previously in panel D have been incorporated into panel E as suggested. The figure is much clearer now we believe.

- The model building for the region of RcpA present in the TadD groove was obviously very challenging, and the authors go to great lengths to present a suitable model, verified experimentally. However, for clarity panel 6a should be moved to figure 7. A close-up view of the density would really help convince us that the model is credible, and the local resolution should be cited. Did the authors try to generate a model with AlphaFold multimer?

We understand the reviewer concerns. We think that panel 6A is in the right place given this figure both analyses the TadD model whilst also presenting the TadD/RcpA interface in detail. We are therefore reluctant to move it to Figure 7 please. However, to address the reviewer's concerns over the credibility of the model we have generated an additional figure (Supplementary Figure 7) where we show detailed zoom boxes of the model build within the map as requested. This figure also serves to help the reader understand how the TadD build was undertaken within the focussed refinement map ($TadD_{FRmap}$) and is referenced where relevant in the main text and Methods. The local resolution for $TadD_{FRmap}$ is shown in Supplementary Figure 5. We did generate a model with AlphaFold multimer and this closely supports our experimental data and structure (Figure 1 below).

Figure 1. TadD-RcpA AlphaFold versus experimental model fitted within $TadD_{FRmap}$. Sequences of TadD and RcpA were modelled by AlphaFold and superposed onto the experimentally derived TadD structure as shown in Supplementary Figure 7. For clarity only RcpAaa412-416 are shown here from the AlphaFold model.

- Figure 8: TadZ should be included (with the previously-determined crystal structure of its cytoplasmic domain). Also, RcpC is usually localized to the OM, not the IM as depicted here. I am aware of the discussions in the community that it may not be the case, but the representation chosen here should be justified (also applies to lines 380-381).

TadZ has been included in the figure. Justification for RcpC depiction within the inner membrane has been outlined in the legend for Figure 8. Specifically, that Clock et al. J Bac 2008 show that RcpC localises to both inner and outer membrane.

Some additional minor comments:

- I think table S1 should be altered: Currently it looks like there are two refined models, one obtained with each dataset. However, there is only one atomic model deposited, which (if I understand correctly) was obtained by merging the two datasets. This needs to be altered accordingly.

Table S1 has been amended and clarified so it is clear there is only one refined and deposited model.

- Figure S3a: The RcpA band appears at the size of a monomer, when co-expressed with TadD. This is contrary to the results reported above, where it forms SDS-resistant oligomers. This needs to be explained. Was this sample boiled, unlike the previous ones?

We cannot be conclusive as to why we do not see SDS-resistant oligomer for the RcpA-mNeon/TadD-mScarlet strain- like all others, the sample was not boiled. We conclude it is likely due to poor membrane transfer or increased sensitivity to SDS denaturation for the RcpA-mNeon construct rather than lack of secretin formation. This is due to i) validation that RcpA-mNeon/TadD-mScarlet constructs do assemble secretin in *E. coli*; ii) RcpA-mNeon does not localise to the cell pole as would be expected if it was not co-localising with TadD. To acknowledge this we have added the following sentence to the figure legend in Supplementary Figure 3: For RcpA-mNeon/TadD-mScarlet we did not readily observe SDS-resistant oligomer consistent with secretin likely due to poor membrane transfer or increased sensitivity to SDS denaturation.

- In figure S4, the set of particles with C14 symmetry is not show. It would be helpful to show this here.

Two representative C14 class averages have been added to Supplementary Figure 4.

- Figure S5a: the dotted line for the resolution criterion is missing.

The dotted line has been added.

- This is not critical here, but did the authors make a model of the RcpA N-domain 13/14-mer, with AlphaFold multimer or using the VirB structure?

The RcpA N-domain is an Alphafold model and its orientation is modelled based on the *X. citri* VirB9 NTD periplasmic ring³⁹. This has now been clarified in the legend of Figure 8.

Reviewer #2 (Remarks to the Author):

TadSS and secretin family systems are extremely important virulence factors that allow bacterial pathogens to cause disease, persist, and to share DNA. Therefore, how they are assembled is not only a fundamentally interesting question, but is also important to solve to help progress the development of novel antimicrobial.

Here Tassinari et al conduct well executed experiments to uncover the structure of the outer membrane complex of the *P. aeruginosa* TadSS – RcpA-TadD – to high resolution. The methods are

appropriate and it was excellent to see the authors going the extra mile to conduct focused classification to gain extra insight into the TadD structure to aid the modelling. Coupling CryoEM with both light-microscopy and crosslinking techniques also made for a thorough analysis accompanied by the appropriate controls to support the conclusions. Overall, the major novel discoveries are that the channel can exist in two different symmetries and the discovery of the RcpA-TadD binding interface. I only have some minor thoughts and minor points for the authors to consider:

Another TPR containing lipoprotein is BamD and it has been shown that it is required for the assembly of the essential BamA outer membrane barrel protein. Similarly, BamD may bind to beta sheets during the folding of barrel protein substrates. It is interesting to see the potential overlap in function of TadD and BamD in their ability to help assemble RcpA. Perhaps there is an evolutionarily conserved functional overlap?

Thank you for this insight of which we were unaware. An additional sentence with references have been included in the discussion: 'It serves as an additional paradigm where TPR-containing proteins, reminiscent of BamD in the BAM complex^{57,58}, function as key factors in the assembly and stabilisation of outer membrane complexes.'

1) It is not clear what the gel in Fig 4B is (Coomassie? Blot?). Also, why is there no RcpA present?

The legend for Fig 4B was not clear. We have now clarified that this was a Western blot using anti-strep antibody against TadD_{strep} hence a band for RcpA should not be expected. Western blot analysis was used as native expression levels were so low standard Coomassie staining of the gel was not useful. This point has now also been clarified in Methods in the 'Native RcpA-TadD_{strep} purification from *P. aeruginosa* PAO1' section.

2) Line 298: "...with sulphur atoms (predicted to be) 4.9A ..."

This has been amended accordingly.

3) Line 308 to 311 and Fig 7E: There is also a corresponding increase in the SDS resistant fraction of RcpA secretin TadD high molecular weight bands in the paired cys samples when CuP treated that might be worth mentioning. (Seemingly, the disulfide bond stabilises the assembled form which goes along well with the conclusions.)

This insight has been included in the text L321. Specifically 'SDS-resistant sample also accumulated in the gel well consistent with increased RcpA secretin assembly'.

(4) The Discussion is very long. Maybe consider condensing.

The discussion has been shortened by removing the paragraph describing secretin structure (formerly paragraph 2). Some of this paragraph has now been integrated into the introduction given details of secretin ultrastructure were missing and as requested by reviewer 1. The text has been further streamlined with grammar improvements and text deletions where possible.